# Use of Cactus Pear Meal in the Feeding of Laying Hens in Semi-Intensive System

**DOI:** 10.3390/ani14040625

**Published:** 2024-02-16

**Authors:** Iara S. Sousa, Roseane M. Bezerra, Edson C. Silva Filho, Leilson R. Bezerra, Ricardo L. Edvan, Stelio B. P. Lima, Elainy Cristina Lopes, Francisca Luana A. Carvalho, Francinete A. S. Moura, Gabriela I. Souza, Leilane R. B. Dourado

**Affiliations:** 1Department of Animal Science, Federal University of Piaui, Teresina 64000-900, PI, Brazil; iarasowzza@hotmail.com (I.S.S.); edvan@ufpi.edu.br (R.L.E.); luanaraielly@hotmail.com (F.L.A.C.); gabriela.jantorno@hotmail.com (G.I.S.); 2Center for Agricultural Sciences, Federal University of Piaui, Bom Jesus 64900-000, PI, Brazil; roseanemadeira@hotmail.com (R.M.B.); steliolima@ufpi.edu.br (S.B.P.L.); francinesett@gmail.com (F.A.S.M.); 3Interdisciplinary Laboratory of Advanced Materials, Chemistry Department, Federal University of Piaui, Teresina 64049-550, PI, Brazil; edsonfilho@ufpi.edu.br; 4Graduate Program in Animal Science and Health, Federal University of Campina Grande, Patos 58708-110, PB, Brazil; leilson@ufpi.edu.br; 5Commercial Consultant, EPE Produtos Agropecuários, Garanhuns 01311-300, PE, Brazil; elainy.clopes@gmail.com

**Keywords:** alternative feed, productive performance, laying, egg quality

## Abstract

**Simple Summary:**

This study explores the effectiveness of the use of cactus pear meal in the feeding of laying hens to improve production and reduce costs. The objective was to evaluate the impact of different proportions of this flour on the birds’ diet. The variety “Miúda” of cactus pear was found to be the most efficient, improving energy utilization and nutrient digestibility. Although the birds’ productive performance was not significantly affected by the use of up to 9% of flour in the diet, improvements were observed in egg quality, including in the texture and color of the yolk, as well as a healthier composition in terms of fatty acids and cholesterol. The study concludes that a 3% inclusion of cactus pear meal is the most economically suitable, offering benefits without compromising quality or performance. These findings are important for egg farmers who seek sustainable and economical alternatives, thus contributing to a more conscious society regarding food production and nutrition.

**Abstract:**

Little information is available in the literature on the use of cactus pear meal (CPM) in poultry diets; therefore, it is important to evaluate diets that provide excellent performance and lower production costs. Our objective was to study the use of Miúda CPM in the diets of laying hens. In the first study, two diets for male and female chicks were used—1: 80% reference diet + 20% Miúda cactus pear meal (CPM) and 2: 80% reference diet + 20% Gigante cactus pear meal (CPM). The variety Miúda provided a better use of metabolizable energy, as well as a greater digestibility coefficient of dry matter, protein, and mineral matter. In the second study, a control diet was compared to three diets with different levels of Miúda CPM for laying hens in the proportions of 3%, 6%, and 9%. No significant differences were found in productive performance. However, there were significant differences in the some parameters egg quality, texture and color profile of the cooked yolk, egg composition, fatty acids and cholesterol in the yolk. It is possible to use 9% Miúda CPM in the diet of laying hens in a semi-intensive system that does not compromise performance and egg quality, and using 3% Miúda CPM provides a higher economic return.

## 1. Introduction

The constantly growing poultry sector faces daily challenges in meeting the nutritional needs of laying hens and maintaining sustainable egg production [1]. One of the main challenges is the high cost of commodities, especially corn, which drives the search for viable feed alternatives to reduce production costs and enrich the birds’ diet. In the midst of these challenges, Brazil, as the world’s fifth largest producer, registered in 2022 a production of 52 billion eggs and an average of 241 eggs consumed per person, highlighting the need for innovative solutions in poultry nutrition [2]. At the same time, there is a growing consumer trend for eggs from sources that provide functional properties, such as antioxidant properties, for health benefits and the prevention of diseases, including chronic diseases, such as coronary heart disease. This trend highlights the importance of exploring innovative and sustainable food alternatives, combining the needs of the industry and consumers [3].

Cactus pear, from the Cactaceae family, comprises around 130 genera and 1500 species, with *Opuntia* and *Nopalea* being the most prominent genera in animal feeding. In Brazil, the species *Opuntia fícus-indica* (L.) Mill with the cultivars Gigante and Redonda and *Nopalea cochenillifera* (L.) Salm-Dyck with the cultivar Miúda are widely cultivated, covering around 600,000 ha [4]. This extensive cultivation, particularly in northeastern Brazil, is due to its adaptability to the region’s soil and climate conditions and its high concentration of non-fiber carbohydrates, offering an efficient and viable source of energy for animal nutrition [5,6,7]. Among the cactus pear cultivars, Miúda (*Nopalea cochinilifera* (L.) Salm Dyck) stands out for its nutritional qualities: high levels of total carbohydrates (822.1 g kg^−1^), non-fiber carbohydrates (597.5 g kg^−1^), mineral matter (128.8 g kg^−1^) and low levels of neutral detergent fiber (224.6 g kg^−1^) and acid detergent fiber (189.7 g kg^−1^) [6], as well as high energy content (3653 kcal g^−1^) [8].

The Miúda cactus pear meal (CPM), obtained by dehydrating the cladodes, has 82.2% dry matter (DM), 8% crude protein (CP), 1% ether extract (EE), 25.1% neutral detergent fiber (NDF), and 46.7% non-fiber carbohydrates (NFC). It is rich in minerals, with 18.5% mineral matter, including 2.3% calcium and 0.2% phosphorus [9]. In addition, with an energy value of 3647 kcal/g gross energy (GE) [8], Miúda CPM stands out for its energy efficiency. In addition, the presence of bioactive compounds such as polyphenols (207.92 mg 100 g^−1^ of total soluble phenols, 647.99 mg 100 g^−1^ of hydrolysable polyphenols, and 3.55 mg g^−1^ of condensed tannins) and β-carotene (4.36 mg 100g ^−1^) endorse it as a promising source of natural antioxidants. The antioxidant activity is significant, with values of 15.28 mmol DPPH, 20.97 mmol for FRAP, and 51.31 mmol for ABTS+, offering protection against free radicals and promoting health benefits for consumers [10].

Although previous studies explored the promising results of CPM in broilers and quails [8,11,12], the literature still needs research into its applicability in laying hen diets. Therefore, this research aims to evaluate the potential of using Miúda CPM to feed layers in a semi-intensive system, with an emphasis on egg production and quality. By investigating this sustainable alternative, we hope to contribute to new perspectives for optimizing egg production, combining the sector’s economic interests with new scientific advances and consumer market expectations in terms of nutritional quality and sustainability.

## 2. Materials and Methods

Two experiments were carried out according to research ethics protocols from Resolution 879/08 of the National Council for Control and Animal Experimentation (CONCEA), with Protocol number 592/19, approved by the Animal Experimentation Committee of the Federal University of Piauí (CEEA—UFPI).

### 2.1. Obtention and Production of the Cactus Pear Meal (CPM)

The cladodes of cactus pear Miúda (*Nopalea cochinilifera* (L.) Salm Dyck) and Gigante (*Opuntia ficus-indica* (L.) Mill) were harvested at the agrostology experimental station of the Federal University of Piauí (UFPI), Bom Jesus, Brazil. Two-year-old cladodes were harvested and transported to the Animal Nutrition Laboratory (LANA) of UFPI, where they were cut (5 to 3 cm thick) and distributed in trays in several layers. The material was pre-dried in oven of forced ventilation at 55 °C for 72 h. The pre-dried material was weighed on an analytical scale and ground in a Thomas Wiley SP-32 SLAPOR^®^ mill in 1 mm mesh sieve; then, it was stored.

### 2.2. Experiment I: Digestibility Trial of the CPM Varieties

#### 2.2.1. Location and Experimental Conditions

The first experiment was carried out in the experimental poultry aviary of the Technical College of Bom Jesus (CTBJ) of the Federal University of Piauí (UFPI) at the Professora Cinobelina Elvas campus in August 2019. The town is located at latitude 9°4′27″ South, longitude 44°21′30″ West, and 277 m altitude. The climate of the region is classified as “Aw” according to the Köppen (1928) classification [13]. The aviary had the sides closed by screens and was provided with mechanized external curtains and fans. The space for metabolic trials had four batteries of nine cages each, measuring 1.16 m × 1.10 m × 0.50 m, with screened floors, feeders and trough-type drinkers, and a removable bottom tray for collecting excreta.

#### 2.2.2. Birds and Experimental Period

A total of 72 male and female Isa Label broiler chicks were used, and the treatments were distributed in a completely randomized design, with three diets and six replications of four birds each. The experimental period ranged from 24 to 32 days of age, with 4 days of adaptation to the diets and 4 days of total collection of excreta. The total excreta collection method was used according to the procedures described by [14]. Before the beginning of the adaptation period, from 1 to 23 days of age, the birds were kept in separate batteries and fed a diet to meet the nutritional requirements established by [15].

#### 2.2.3. Diets and Analyses

The treatments consisted of a reference diet (RD) based on corn and soybean meal (Table 1) and two test diets containing two varieties of cactus pear: Test Diet 1, which included 80% RD and 20% Miúda cactus pear meal (CPM), and Test Diet 2, containing 80% RD and 20% Gigante cactus pear meal (CPM).

Then, 1% iron oxide was added to all diets to serve as a marker for the beginning and end of the collection period. Collections were made twice daily in trays covered with plastic material. After collection, the excreta were weighed, packed, identified, and frozen at 18 °C until the end of the collection period, when they were thawed, weighed, and homogenized; then, a sample was removed, weighed, and dried in forced ventilation oven at 65 °C for 72 h. The oven-dried samples were weighed, ground, and stored for laboratory analysis.

The contents of dry matter (DM), mineral matter (MM), crude protein (CP), and gross energy (GE) were determined in all samples following the methods of [16]. With the laboratory results, the apparent metabolizable energy (AME), dry matter digestibility coefficient (DMDC), crude protein digestibility coefficient (CPDC), and mineral matter digestibility coefficient (MMDC) were determined according to [14].

The results obtained from the analyses for ingested and excreted diets were used to determine the apparent metabolizable energy corrected for nitrogen balance (AMEn) of the experimental diets [17]. The same principle was used to calculate the DMDC, CPDC, and MMDC [17].
$$Nutrient\;digestibility\;coefficient\;(NDC)RefB=\frac{\left(Nutrient\;ingested-Nutrient\;excreted\right)\times 100}{Nutrient\;eingested}$$
$$NDC\left(Cactus\;Pear-CAP\right)=NDC\left(Ref\right)+\frac{(NDC\left(diet\;with\;CAP\right)-NDC\left(Ref\right)}{\%CAP\;substitution}$$

### 2.3. Experiment II: Production Performance and Egg Quality

#### 2.3.1. Location and Experimental Conditions

The second experiment was carried out at the School Farm Alvorada do Gurguéia (FEAG) of the Federal University of Piauí, located in the town of Alvorada do Gurguéia, Piauí, Brazil, from August to October 2020. The town is in the Upper Middle Gurguéia microregion located at the following geographic coordinates: latitude 8°22′34″ South, longitude 43°51′23″ West, and 220 m of altitude. The climate of the region is classified as “Aw” according to the Köppen (1928) classification [13]

The laying hens were housed in a brick aviary with clay tile roof in 20 boxes measuring 2 m × 1 m. Each box had access to an external paddock of 9.80 m × 5.45 m, surrounded by wire (Figure 1). The boxes were equipped with side curtains with manual adjustment system, bedding of rice straw, a nest of 30 cm × 35 cm covered with rice straw, a tubular feeder, and a pendular drinking trough.

#### 2.3.2. Birds and Experimental Period

Initially, 80 one-day-old female chicks of the Bankiva GLK strain were housed in prepared boxes and fed a diet based on corn and soybean meal, according to the guidelines of the Embrapa Brown Egg Colonial Laying Hen Management Guide [18]. The life stages were categorized as brood (1st to 6th week), rearing (7th to 15th week), and pre-laying (16th to 31st week). During the pre-laying phase, egg production was monitored until approximately 80% laying was achieved, marking the start of the experimental phase. At 31 weeks of age, the hens were weighed for plot assembly, according to the experimental design, and housed for adaptation to new groups. At 32 weeks of age, hens were reweighed, presenting an average initial weight of 1.647 kg, and distributed in 20 boxes of similar weights. At the end of the experimental period, at 40 weeks of age, the hens were weighed again, presenting an average final weight of 1.679 kg. The treatments were distributed in a completely randomized design with four treatments and five replications of four hens each. The study lasted for 63 days, divided into three periods of 21 days each.

#### 2.3.3. Diets and Management

The experimental diets (Table 2) were formulated according to the nutritional requirements suggested by the management recommendation guide for free-range laying hens (32th to 40th week) of brown eggs [18]. Nutritional values of ingredients are according to [19], except for the cactus pear, which considered the composition of calcium, phosphorus, sodium, potassium, NDF, and ADF determined by [6]. The total amino acid composition was determined by [20], and the digestible amino acids by the digestibility coefficient of feeds with fiber content similar to cactus pear according to [19] (Table 3). The cactus pear used in this experiment was the Miúda (*Nopalea cochinilifera* (L.) Salm Dyck) from a macro project that prioritized saving water through plant fertilization using hydrogels [21,22]. The composition and metabolizable energy of cactus pear (Miúda) analyzed in the first experiment were considered for the formulation of the experimental diets to maintain the same nutrient levels in all treatments.

The diets consisted of levels of Miúda CPM: Treatment 1: control diet based on corn and soybean meal; treatments 2, 3, and 4: diet with 3%, 6%, and 9% of Miúda CPM, respectively.

Specific practices were adopted to stimulate feed intake and maintain hygiene in the layers’ environment. Feed and water were offered freely during the experimental period. The feed was stirred twice a day to stimulate feed intake, and the drinking troughs were cleaned daily to prevent the accumulation of dirt and maintain water quality. These management practices ensured the birds’ well-being.

Eggs were collected manually at 1 p.m. and 5 p.m. The first collection was after morning laying when most of the hens already laid. The second collection included cleaning and closing the nests and reopening them the following morning to maintain hygiene. All eggs collected were cleaned and stored at room temperature. The side curtains were handled manually, being raised in the morning and lowered in the late afternoon.

Lighting was established according to the age of the hens, as well as the sunrise and sunset times of the region, using a photoperiod of 14 h of light and 10 h of dark, with 12 h of natural light and 2 h of artificial light. The light supply was operated manually, with half being offered in the morning (5 to 6 a.m.) and the other half in the evening (6 to 7 p.m.). Air temperature and humidity were monitored by two digital maximum and minimum thermo-hygrometers TOMATE^®^ (Model PD003, TOMATE, São Paulo, Brazil) located inside two boxes on opposite sides at the height of the hens’ backs. Climatic data were collected at 6 a.m. throughout the experimental period.

#### 2.3.4. Productive Performance

The following performance parameters were evaluated: feed intake (FI) (g/bird/day), water intake (WI) (mL/bird/day), egg production (EP) (%), egg weight (EW) (g), egg mass (EM) (g), conversion per egg mass (CEM) (kg/kg), and conversion per egg dozen (CED) (kg/dozen). The variables were analyzed in three cycles of 21 days each. The analyses occurred in the 34th, 37th, and 40th week (17 days after the beginning experimental feed supply). Feed intake was calculated by the difference between the amount of feed provided and the experimental leftovers, weighed at the beginning and end of each 21-day period. Water intake was calculated by measuring the supplied and leftover water for four consecutive days of each period. Egg production was recorded during the three 21-day periods on laying sheets twice daily and obtained by dividing the sum of eggs per period and per plot by the number of birds and then multiplying by 100. On the last four days of each period, eggs from each plot were weighed individually to obtain the average egg weight. Egg mass was calculated as the product of egg production and average egg weight per plot. Feed conversion per egg mass was calculated by the ratio between feed intake and egg mass produced, and the conversion per egg dozen was calculated by the ratio between feed intake and production in dozens.

#### 2.3.5. Egg Quality

Variables were analyzed in three cycles of 21 days each. The analyses occurred in the 34th, 37th, and 40th weeks (17 days after the beginning of the experimental feed supply). Quality variables were evaluated in three eggs per plot, from 18 to 21 days of each period, obtaining the following parameters: yolk diameter (YD) (mm), which was determined by a Pantec^®^ digital pachymeter (Model 150MM/6, Pantec, São Paulo, Brazil); yolk percentage (YOLK), albumen percentage (ALB), and shell percentage (SHELL), which were obtained by dividing the component weight by the egg weight and multiplying the result by 100; shell thickness (ST) (mm), which was measured in three parts (apical, equatorial, and basal) using a digital pachymeter; specific weight (SW) (g/cm^3^), which was performed by the salt flotation method as described by [23], where the eggs were immersed in salt solutions with densities ranging from 1.070 to 1.090 with an interval of 0.0025, with densities adjusted using petroleum densimeter; and Haugh Unit (HU), which was determined using the equation: HU = 100 log (H + 7.57 − 1.7W0.37), where H = albumen height (mm); and W = egg weight (g) [24,25].

On the last collection day of each period, two eggs per plot were analyzed for Shell Resistance (SR) and Yolk Resistance (YR) (kgf) through the Texture Analyzer Brookfield^®^ (model CT3 50 kg), connected to a computer equipped with TexturePro CT software, V1.9 Build 35. Three color parameters were evaluated: L*, a*, and b*. The a* value characterizes coloration in the red (+a*) to green (−a*) region, and the b* value indicates coloration in the yellow (+b*) to blue (−b*) range. The L* value gives the brightness, ranging from white (L = 100) to black (L = 0) [26]. The coloration parameters were measured at three different points on the egg yolk with the aid of a Minolta CR-400^®^ (KONICA MINOLTA, Inc., São Paulo, Brazil) portable colorimeter in the CIELab system.

#### 2.3.6. Analysis of Egg Texture Profile and Color of the Cooked Yolk

Eggs from each treatment were used for egg texture profile analysis, in which the variables analyzed were hardness, cohesiveness, elasticity, gumminess, whole egg chewiness, and yolk color parameters. Eggs were cooked by treatment for 10 min and placed in cold water for 3 min to be peeled. Whole egg texture was assessed using the Texture Analyzer Brookfield^®^ (model CT3 50 kg), connected to a computer equipped with TextureLoader^®^ software. The speed of the slide test was 2.00 mm/s in 2 cycles. After that, the yolk was separated manually to be analyzed for coloration by a Minolta CR-400^®^ portable colorimeter in the CIELab system.

#### 2.3.7. Egg Composition, Fatty Acids and Cholesterol in the Yolk

Cooked yolk and albumen samples were frozen after texture profile analysis. Then they were thawed, weighed, and prepared to be freeze-dried in for 72 h, then ground in a small domestic mixer to a powder. The freeze-dried powdered samples were analyzed for DM, CP, and MM content through the methods of [16].

The fatty acids (FAs) in the samples were methylated according to [27]. The resulting fatty acid methyl ester was determined using a gas chromatograph (model Focus GC; Thermo Scientific, Milan, Italy), equipped with a flame ionization detector and SP 2560 fused silica capillary column (100 m × 25 mm × 0.2 µm film thickness; Supelco, Bellefonte, Pennsylvania). Hydrogen was used as carrier gas (1 mL/min), and nitrogen as an auxiliary gas. The detector and injector temperatures were set at 250 °C, with a split ratio of 15:1. The oven temperature was set to 70 °C for 4 min, increased by 13 °C/min until 175 °C, held for 27 min, then increased by 4 °C/min until 215 °C, and held for 31 min. FAMEs were identified by comparing three FAME references (Supelco FAME mix C4-C24, CLA trans-9, cis 11 16413, and CLA trans-10, cis 12 04397; Sigma Aldrich, St. Louis, MI, USA).

Quantification of total cholesterol (TC) in yolk from the 32nd to the 40th week of age was performed according to the methodology described by [28]. Saponification of 0.5 mL of lipid extract was performed in a 50-mL Falcon tube, adding 10 mL of potassium hydroxide solution (KOH 2%) in 90% ethanol. Then the tubes were placed in water bath at 80 °C under stirring for 15 min. Subsequently, 5 mL of distilled water was added and allowed to cool. For extraction of unsaponifiable matter, 10 mL of hexane was added, stirring in vortex for 1 min. After separation, the entire hexane phase was transferred to another Falcon tube, and the extraction was repeated two more times. About 4 mL of hexanoic extract was collected and evaporated in a water bath at 55 °C, added 6 mL of saturated acetic acid in iron sulfate, cooled in a gel bath, and vortexed for 1 min. Immediately after, about 2 mL of sulfuric acid was added and cooled to 26 °C. After 10 min, the reading was taken in a spectrophotometer at 490 nm. Different concentrations from 0 to 200 ppm of purified cholesterol (sigma) were used in the standard cholesterol curve, and the absorbances were performed in a UV-VIS^®^ spectrophotometer (Thermo Scientific, Milan, Italy).

#### 2.3.8. Economic Viability

The economic viability analysis was performed through the price of feed per kilogram (kg, BRL), feeding cost per dozen eggs (FCDZ, BRL/each), and relative gross margin (RGM, %), that is, the gross margin (GM, BRL) of the treatments using Miúda CPM in comparison to the GM of the control treatment. The determination of RGM was performed according to [29], considering only the variable costs of feeding, since the fixed costs were the same for all treatments.

The input prices (BRL/kg) used for the price calculations of the kilogram of feed were as follows: corn = BRL 1.10; soybean meal = BRL 3.70; wheat bran = BRL 1.30; soybean oil = BRL 5.56; dicalcium phosphate = BRL 5.40; limestone = BRL 4.00, common salt = BRL 0.80; Vitamin supplement = BRL 6.40; DL-methionine = BRL 15.00; L-lysine HCL = BRL 7.00; L-threonine = BRL 8.00; L-tryptophan = BRL 60.00, L-arginine = BRL 95.00; and Miúda CPM = BRL 0.20. To calculate the price of CPM, it was taken into consideration only the expenses of handling the product (labor). The price of feed per kilogram (BRL/kg) was obtained by multiplying the price by the quantity of ingredients used per treatment/100. The feeding cost per dozen eggs (FCDZ, BRL/dozen) was obtained by the price of feed kilogram and bird intake divided by 12. The price per dozen eggs (PDZ, BRL/per dozen) was BRL 8.00 in the local of Bom Jesus-PI at the end of the experiment (October 2020).

### 2.4. Statistical Analyses

Data of the variables obtained were subjected to univariate analysis of variance using PROC GGLM, and compared by SNK test at 5% probability. The estimates of the use of cactus pear were established by polynomial regression for the significant variables. The SAS^®^ OnDemand for Academics was used for the analyses (https://www.sas.com/pt_br/software/on-demand-for-academics.html). The results figures were prepared in the Microsoft Office Excel Program. Autodesk AutoCAD, version 24.3.61.0 (Autodesk, Inc., San Francisco, CA, USA), was used to illustrate the experimental area.

## 3. Results

### 3.1. Metabolizable Energy and Digestibility Coefficient of CPM

The contents of 3549 and 3399 kcal kg^−1^ GE and 1402 and 1142 kcal kg^−1^ AME were found for the varieties of Miúda cactus pear meal (CPM) and Gigante cactus pear meal (CPM), respectively, for chicks aged 24 to 32 days. The GE values are the result of triplicate analysis of each cactus sample and no statistics were performed. There was no statistically significant difference in metabolizable energy (Figure 2). A statistically significant difference in DMARD was observed between the Miúda CPM and Gigante CPM varieties. However, no significant statistical differences were identified in CPDC and MMDC between the Miúda CPM and Gigante CPM varieties (Figure 3).

### 3.2. Climate Data

Temperature and air relative humidity averages for the entire experimental phase of the second experiment are presented in Table 4. The relationship between temperature and nutrition should be analyzed and taken into consideration in the rearing of commercial laying hens, since the variation in ambient temperature regulates mainly feed intake. Temperature did not affect the production and egg quality data.

### 3.3. Productive Performance

The diets containing levels of Miúda CPM did not promote significant effects on the variables egg production, feed intake, water intake, egg weight, egg mass, conversion per egg mass and conversion per egg dozens of laying hens in semi-intensive system from the 32nd to the 40th week of age (Table 5).

### 3.4. Egg Quality

The effects of Miúda CPM levels on egg quality parameters of laying hens reared in semi-intensive system are presented in Table 6. There was a linear reduction between treatments in yolk diameter (YD = 34.14 − 0.0991CPM, R^2^ = 0.9864) that is, yolk diameter reduced linearly as the level of Miúda CPM increased in the diet. Shell resistance (SR = 4363.94 − 50.9031CPM, R^2^ = 0.9178), yellow coloration (b* = 58.35 − 0.8404CPM, R^2^ = 0.8853), and red coloration (a* = 9.68 − 0.3897CPM, R^2^ = 0.9707) also decreased linearly as the level of Miúda CPM increased. On the other hand, there was a linear increase in albumen percentage (ALB = 62.00 + 0.0881CPM, R^2^ = 0.90) as the levels of Miúda CPM increased in the diet. However, there was no significant effect on yolk percentage, shell percentage, shell thickness, specific weight, Haugh Unit, yolk strength, and yolk brightness with increasing levels of Miúda CPM.

### 3.5. Analysis of Egg Texture Profile and Color of the Cooked Yolk

The effects of Miúda CPM levels on the parameters of egg texture profile and yolk color of cooked eggs are presented in Table 7. There was a linear increase in cohesiveness (COHE = 0.384 + 0.0085CPM, R^2^ = 0.16), which means that as the level of Miúda CPM increased, the cohesiveness of the cooked egg also increased. The red coloration of the yolk reduced linearly (a = 6.405 − 0.305CPM, R^2^ = 0.29) as the level of Miúda CPM increased. In contrast, for the yellow coloration of the yolk, there was a quadratic effect (b = 41.965 − 2.917CPM + 0.223CPM^2^, R^2^ = 0.54), with the color b reducing up to 6.52% with the use of Miúda CPM.

### 3.6. Egg Composition, Fatty Acids, and Cholesterol in the Yolk

The effects of levels of Miúda CPM on egg chemical composition, fatty acids, and total cholesterol in the yolk are presented in Table 8. It was found to have a decreasing effect on albumen MM (MM_Albumen_ = 2.699 − 0.1591 CPM + 0.0134 CPM^2^, R^2^ = 0.75), while there were no effects on albumen DM and CP. There were also no significant effects on the yolk DM, MM, and CP of the increasing levels of Miúda CPM. The use of Miúda CPM levels did not promote a significant effect on total yolk FAs but linearly reduced myristic = 0.4758 − 0.01002 CPM, R^2^ = 0.5, Myristolic = 0.0802 − 0.00126 CPM, R^2^ = 0.68, and oleic = 39.5211 − 0.1828 CPM, R^2^ = 0.61. There was also a reduction in palmitic = 27.423 − 0.5864 CPM + 0.0386 CPM^2^, R^2^ = 0.84, palmitoleic = 4.2453 − 0.4056 CPM + 0.0274 CPM^2^, R^2^= 0.98, and vaccenic = 2.223 − 0.0939 CPM + 0.00462 CPM^2^, R^2^ = 0.98. Cactus pear levels promoted linear increase in α-linolenic = 0.4163 + 0.0253 CPM, R^2^ = 0.84, and other FAs = 4.598 + 0.0573 CPM, R^2^ = 0.59, as well as increase in stearic = 7.575 + 0.2237 CPM − 0.0196 CPM^2^, R^2^ = 0.68, linoleic = 12.368 + 0.8296 CPM − 0.03731 CPM^2^, R^2^ = 0.99, and Arachidonic = 1.7061 + 0.0523 CPM − 0.0046 CPM^2^, R^2^ = 0.86. The total cholesterol in the yolk of laying hens fed CPM in a semi-intensive system also increased linearly (TC = 397.4 + 20.37 CPM, R^2^ = 0.77).

### 3.7. Economic Viability

Economic viability data are presented in Table 9. Feed intake was greater for birds receiving the treatment with 6% Miúda CPM, while the treatment with 9% Miúda CPM had the lowest intake. The price for the formulation of the diet with 9% Miúda CPM was higher. Feeding cost, egg dozen, and cost per egg dozen were higher for the treatment with 6% Miúda CPM when compared to the other treatments. The price per dozen eggs was BRL 8.00 according to the region and season of the experiment. The gross income was higher when using the treatment with 6% Miúda CPM, while the gross margin was higher for the treatment with 3% Miúda CPM.

## 4. Discussion

The contents of 3549 and 3399 kcal kg^−1^ of GE and 1402 and 1142 kcal kg^−1^ AME (Figure 2) were found in the present study for the varieties of Miúda cactus pear meal (CPM) and Gigante cactus pear meal (CPM). The study of [30] found values of 4009, 3757, and 3945 kcal kg^−1^ GE; 3144, 3019, and 1624 kcal kg^−1^ AME for corn, sorghum, and wheat bran, respectively, were reported for chicks aged 26 to 33 days. The authors [31] found new gross energy values in corn grain (3884 kcal kg^−1^), wheat grain (3867 kcal kg^−1^), and sorghum grain (3987 kcal kg^−1^), while for the metabolizable energy, values of 3719 kcal kg^−1^ for corn, 3265 kcal kg^−1^ for wheat, and 3695 kcal kg^−1^ for sorghum were found for chicks aged 22 to 28 days. At the same time, [32] found metabolizable energy values of 1259 kcal kg^−1^ and 1316 kcal kg^−1^ for corn and wheat, respectively, for 7-, 14-, 21-, 28-, and 35-day-old birds. However, these differences in gross energy and metabolizable energy of those feedstuffs were expected since there are variations in soil conditions, climate, raw material obtention, storage time, processing, age of the birds, physiological state, methodology used, and chemical composition [31,33,34].

The AME value of Miúda CPM (1402 kcal kg^−1^) was higher than that of Gigante CPM (1142 kcal kg^−1^). Although not statistically significant, they may be relevant or indicative of trends that deserve further investigation. Such differences may be related to the species and chemical composition of the varieties of cactus pear, which may interfere with the metabolizable energy. The AME value of cactus pear Miúda stood out due to its bromatological composition, which presented lower contents of soluble fiber when compared to Gigante [7,35,36].

The authors [37] state that the metabolizable energy is directly and positively affected by the composition of the feed in starch, fat, and protein and negatively affected by the structural carbohydrates of the plant. The AME value of Gigante CPM may have been affected by the soluble fiber content (NSPs and pectin), mainly the high content of pectin and its high water solubility [36,38]. These physicochemical characteristics of the soluble fiber fraction result in the increasing viscosity of the digest. High viscosity decreases the diffusion rate of endogenous enzymes in the digest, which will reduce nutrient digestion. In addition, the highly viscous digest will have less interaction with enzymes in the brush border membrane, which also decreases digestibility and nutrient utilization [38,39].

On the other hand, the AME value of Miúda CPM was positively affected by the high contents of non-fiber carbohydrates, mainly starch, being the main source of energy for birds [7,36,40,41]. These results are attributed to the higher intake of non-fiber carbohydrates, which consequently provided higher energy intake [42].

The DMDC value of Miúda CPM (43.5%) was higher than Gigante’s CPM (27.9%). The reduced dry matter digestibility of the cactus pear varieties, mainly for the Gigante CPM, may be attributed to considerable amounts of NSP that cannot be digested by birds because they lack endogenous enzymes. Soluble NSPs can increase the viscosity of the digestate and reduce nutrient digestibility [43]. The superiority in DMDC of Miúda CPM can be attributed to the sugar and starch contents [7,36] since the concentration of these carbohydrates contributes considerably to high palatability, which explains the higher dry matter digestibility of that variety, corroborating the results represented in Figure 2.

The CPDC values of Miúda CPM (36.0%) and Gigante CPM (25.8%), found in this study, were lower than the values found by [44], who reported a protein digestibility coefficient of 92.69% for corn and 91.41% for sorghum for Isa Label chickens from 28 to 35 days. Moreover, [45] found a protein digestibility coefficient of 75.24% for corn and 87.84% for sorghum for Label Rouge birds. These significant differences in the CPDC of CPM, corn, and sorghum are related to the chemical composition of the feed (antinutritional factors and the amount of fiber), in addition to the strain of the birds that can influence the digestibility of nutrients. The lower CPDC may be because the cactus pears are from the genera *Opuntia* (Gigante) and *Nopalea* (Miúda); that is, the genus influences composition, and the composition influences nutrient utilization. The cactus pear variety Gigante presents a higher concentration of soluble fibers [7,35,36]. Soluble fibers impair protein digestibility because they increase the viscosity of the intestinal contents, reducing the action of proteolytic enzymes and consequently causing endogenous nitrogen losses [46]. Another possible explanation is that lignin is a substance of the insoluble fraction of fiber, and its binding with proteins makes it unavailable for animal absorption [35].

For MMDC, the value found for Miúda CPM in this study was higher than the value found by [47], while the variety Gigante CPM responded inferiorly but very close to the value found of a digestibility coefficient of 27.66 and 21.42% for young and adult Label Rouge birds, respectively, fed with feed based on corn and soybean meal. Cactus pear is considered a good source of minerals, regardless of the species (*Opuntia* and *Nopalea*), with the highest concentrations found for Ca, K, Mg, and P and the lowest for Cu, Fe, Sr, and Zn [48,49]. However, cacti possess the antinutritional factor calcium oxalate, which binds to calcium and possibly other minerals in a nutritionally unavailable form, thus interfering with the bioavailability of calcium for animal absorption [35,50]. The researchers [51] observed that the morphology of calcium oxalate crystals was different since the crystals were larger (ranging from 30 to 100 ÿm) and more abundant in fresh cladode tissues of the three *Opuntia fícus-indica* cultivars (Argelina, Morado e Gymno-carpo) than in *Opuntia* robusta, which were smaller (ranging from 6 ÿm to 35 ÿm), more rounded, very sparse, and observed mainly near the epidermis. This caused a reduction in the calcium concentration in *Opuntia* Robusta. Possibly, the lower mineral matter digestibility coefficient of Gigante CPM (19.2%) may be associated with calcium oxalate crystals.

Diets containing different levels of Miúda CPM usage did not compromise the variables of productive performance despite the presence of NSPs and oxalic acid. Although there are no studies available in the scientific literature on the use of CMP for laying hens, various research works point to alternatives to corn for these birds, but with problems that limit their use. For example, rice bran, an alternative to corn, contains a high percentage of phytic acid and NSPs [52], making it a feed similar to CPM due to its chemical composition and presence of antinutritional factors. Knowing about the presence of the antinutritional factors in rice bran, [53] tested the inclusion of rice bran in laying hens’ feed and found that it had no significant effect on egg production, feed intake, feed conversion, and egg mass, as did the present study.

This study is important because it is the first to investigate the use of Miúda CPM in laying hens. The genetic variety of the hens, as well as the specific conditions of the semi-intensive system and the duration of the experiment, provided an opportunity to evaluate the effects of using Miúda CPM on egg production. Although this study focused on only two varieties of Miúda CPM, it opens the way for a wide range of future research that can explore other varieties. The finding that Miúda CPM had no adverse effects on the variables studied is a significant step toward understanding the viability of Miúda CPM as a feed alternative in poultry systems. It is essential that further research is carried out to confirm and expand these results, exploring different scenarios and experimental conditions, as well as evaluating parameters such as the general health of the birds to provide a more comprehensive understanding of the effects of Miúda CPM in poultry farming.

Yolk percentage, shell percentage, shell thickness, specific gravity, Haugh Unit, yolk strength, and yolk brightness values had no significant effects from the increasing levels of Miúda CPM. Miúda CPM has a high concentration of non-fiber carbohydrates [9], which makes it a good alternative source to corn; however, there is a limitation of use due to the concentration of NSPs [54]. There are no reports in the scientific environment of its use in laying hens, so it is acceptable to compare results with similar feedstuffs in terms of energy and fiber (NSPs). Wheat bran is widely used for laying hens due to its availability and energy, but it is limited due to the amount of NSPs. They [6] found that hens responded similarly to the present study; that is, there was no significant effect of adding 3 and 6% wheat bran and beet pulp in the diets of 90-week-old laying hens on egg shape index, yolk percentage, shell percentage, shell thickness, Haugh Unit, and specific gravity.

Eggs from birds that received Miúda CPM levels showed lower yolk diameter values. In contrast, those fed a corn and soybean meal-based diet exhibited larger yolk diameters. However, there is a lack of studies exploring this trait with similar feeds. Yolk diameter is an important variable since it is directly related to the reactions that occur in the albumen, where the water from the albumen crosses the yolk membrane by osmosis and is retained in the yolk. Excess water in the yolk determines the increase of its volume, leading to the weakening of the yolk membrane. This makes the yolk appear larger and flattened when the egg is observed on a flat surface after it is broken [55].

The percentage of albumen increased proportionally as the levels of Miúda CPM increased. An opposite behavior was observed between the percentage of albumen and yolk, particularly evident when using 9% Miúda CPM. This inversely proportional relationship manifests itself with an increase in the percentage of albumen accompanied by a decrease in the percentage of yolk. Presumably, this increase in albumen percentage must be related to the linoleic acid in the birds’ diets (Table 2). The experimental diets were formulated to contain the same metabolizable energy, so as the level of cactus pear increased, it was necessary to increase the amount of soybean oil in the feed to standardize the metabolizable energy. Soybean oil has a reasonable amount of linolenic acid [56], and this acid promotes increased concentrations of estrogen, which is important in controlling egg weight since dietary fats influence egg weight [57,58]. The authors [57] found that diets with supplemental fat and linoleic acid increased the albumen weight of eggs of Isa Brown hens from the 22nd to 65th week of age.

Alternative feeds to corn are well explored to reduce the cost of poultry production. Understanding the importance of exploring the effects of these feeds on egg quality, ref. [59] evaluated a combination of alternative ingredients and found that the percentage of albumen was higher in group 4 (64.06) than in the other groups (1–63.24, 2–63.27, and 3–63.56); these values are close to those found in the present study.

Hens fed 9% Miúda CPM had lower shell strength when compared to the eggs of hens receiving the control feed, reflecting a decrease in shell strength as the Miúda CPM level increased. This reduction may be due to the effect of oxalic acid present in the cactus pear since it is an organic compound that binds to calcium or other minerals in an unavailable nutritional form, affecting the availability for absorption by the animal [35,60], thus causing a deficiency of important minerals for the formation of the shell since about 94–95% of the dry eggshell is composed of calcium carbonate (CaCO_3_).

For yolk coloration, brightness had no significant effect among the experimental diets, but hens fed with 9% Miúda CPM had significantly lower values in the red-to-green region and coloration in the yellow-to-blue range. However, the intensity of the yolk color was higher in the control diet, which may be due to the reduced amount of corn in the experimental diets (3%, 6%, and 9%). A possible reason for this result is that corn is the ingredient source of carotenoids in poultry feeds, and these carotenoids are classified into xanthophylls and carotenes [61,62] added 15% almond shell in the feed of laying hens and found a decrease in the values of a (greener) and b (less yellow) in yolk coloration.

The method of texture profile analysis is based on compressing the food at least two times, simulating the action of two bites on the food. There was no significant effect on the conversion indicator, which deals with changes suffered (weight increase or reduction) by the cooking process. The hardness property, which is the force required to achieve a deformation of the sample, had no significant effect. Regarding the cohesiveness property, there was a linear increase according to the levels of Miúda CPM used. Cohesivity is defined as the degree to which a material is deformed before it breaks (physical) or between the teeth before it breaks (sensory) [63]. Probably, this significant effect on cohesiveness is related to the amount of fat present in the yolk (Table 8) since [64] reported that fat in the yolk increases cohesiveness. No significant effect was observed on the elasticity property, which is defined as the degree to which the deformed material returns to its original condition after a force was applied (physical) or pressed between the teeth (sensory). No effects were found for the parameter gumminess either. This is a parameter defined as the energy required to disintegrate a food into a swallowable state. Regarding chewability, which is the number of chews required, at a constant force, for the food to be swallowed [64], there was no significant effect either.

The cooked yolk coloration parameters a* and b* were affected by Miúda CPM levels since as the level of Miúda CPM increased, the yolk color reduced. The reduction in yolk color intensity may be related to the presence of natural pigments (lutein, zeaxanthin, and β-carotene) [65]. Possibly, this variation is because corn is the main carotenoid source in poultry feed [61], which means that the presence of pigments is higher in corn than in cactus pear. Moreover, [66] showed that the average contents of lutein, zeaxanthin, and beta-carotene in green corn kernels are 0.71, 9.85, and 0.88 µg g^−1^ in the fresh sample, respectively.

The mineral matter of eggs from hens fed with Miúda CPM was reduced; however, dry matter and crude protein did not differ between treatments. This possibly occurred because cactus pear has the antinutritional factor calcium oxalate, which is an organic compound that binds to calcium or other minerals in an unavailable nutritional form, affecting availability for animal absorption [35,60].

According to [67], chromatographic analyses of total lipids extracted from cactus pear cladodes show that palmitic acid (C16:0), oleic acid (C18:1), linoleic acid (C18:2), and linolenic acid (C18:3) contribute in 13.87, 11.16, 34.87, and 32.83% of the total fatty acid content, respectively. These four fatty acids, therefore, represent over 90% of the total fatty acids, with linoleic and linolenic acids being the main polyunsaturated ones, totaling 67.7%.

The saturated fatty acids identified were myristic, palmitic, and stearic. As the level of Miúda CPM increased, there was a reduction in myristic and palmitic acids. However, the behavior of stearic acid was inversely proportional: as the level of Miúda CPM increased, the value of stearic in the yolks also increased.

The monounsaturated fatty acids identified were myristoylic, palmitoleic, oleic, and vaccenic. All monounsaturated fatty acids showed higher concentrations in the yolk of eggs from hens fed with the control diet (0% Miúda CPM) and reduced as the level of Miúda CPM increased. Regarding the oleic acid, a possible explanation for the reduction is the presence of soybean oil in the feed since, in the study of [56], the incorporation of soybean oil reduced oleic acid in yolks from chickens fed with corn.

The polyunsaturated fatty acids, linoleic and α-linolenic, increased as the level of Miúda CPM increased, possibly due to the incorporation of soybean oil in the feed. As the level of Miúda CPM increased, the amount of soybean oil also increased. Soybean oil is rich in linoleic acid and has a fair amount of linolenic acid. The inclusion of soybean oil in the diet increased the linoleic and linolenic acid contents and, consequently, increased the linoleic and α-linolenic fatty acids in the yolk [56]. Linoleic acid promotes increased estrogen concentrations and, thus, stimulates protein synthesis in the oviduct, causing greater protein deposition in the albumen, resulting in a heavier egg [58]. In addition, linoleic acid has long been accepted as having a hypocholesterolemia effect and inhibitory properties against metastatic colon cancer cells. Omega-3 linolenic acid is known to be beneficial to health, cardiovascular disease, inflammatory conditions, autoimmune disorders, and diabetes [67]. Arachidonic acid, which is the precursor of linoleic acid, was detected in the yolks, and the lowest concentration was found in eggs from hens receiving the control diet. As the Miúda CPM level increased in the diet, the concentration of this fatty acid also increased.

The cholesterol content in the yolk increased linearly with the use of Miúda CPM (TC = 397.4 + 20.37CPM, R^2^ = 0.77). This can be explained by the increase in the polyunsaturated fatty acids, linoleic and α-linolenic. The lipid composition of egg yolk can be altered, especially regarding the fatty acid profile, including the content of n-3 polyunsaturated fatty acids (PUFA) [68]. The cholesterol content of egg yolks has become an important issue for consumers, especially when it comes to preventing chronic diseases, including coronary heart disease. Cholesterol is synthesized by the human body and consumers have been advised to avoid ingesting cholesterol in the diet to prevent these diseases. More recently, it was determined that exogenous cholesterol actually represents a very small amount of blood cholesterol [69].

Although feed intake was higher for hens receiving the diet with 6% Miúda CPM, the price of feed was higher in the 9% CPM diet in comparison to the others, which may be due to the increase in soybean oil in the diets to keep them isoenergetic. Consequently, feeding cost was also higher for the diet containing 6% CPM due to the higher feed intake. Egg dozen production (dozen/bird) was higher when hens consumed diets containing 6% Miúda CPM. A plausible explanation is that with 6% cactus pear meal in the diet, there was higher feed intake; consequently, more money was spent to produce the 6% Miúda CPM diet. In contrast, the ratio of feeding cost/egg dozen (BRL/dozen) was higher for the control diet, while for the diet with 6% Miúda CPM, the feeding cost to produce a dozen eggs was higher than the diets with 3% and 9%, respectively.

## 5. Conclusions

Although this study opened up new perspectives on the use of Miúda CPM in poultry nutrition, further studies are needed to elucidate its antioxidant potential and impact on poultry products. This indicates a promising direction for future research, where the composition of bioactive compounds in cactus pear and their potential beneficial effects on poultry products can be further explored.

Considering the results, it is possible to use 9% of Miúda CPM in the diet of laying hens in a semi-intensive system with no adverse effects on prejudice to the performance parameters and external and internal quality of eggs. When using 3% of Miúda CPM, a higher economic return was obtained.

## Figures and Tables

**Figure 1 animals-14-00625-f001:**
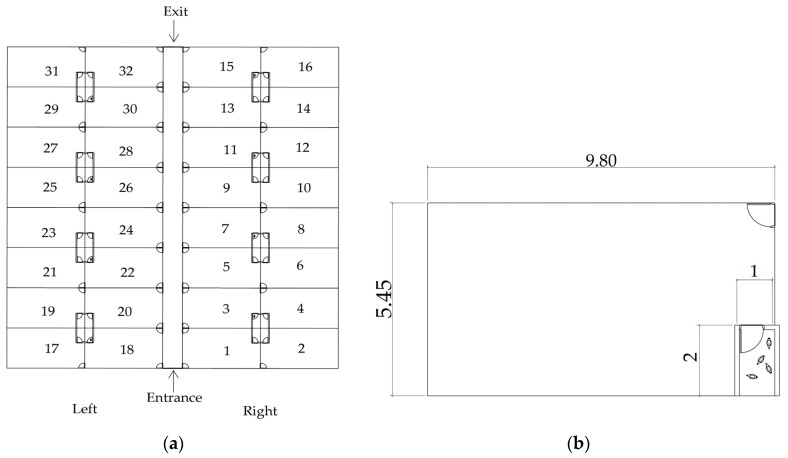
Overview of the experimental area used in the laying hen experiment, showing the layout of the boxes and paddock (**a**); detailed view of an individual box and paddock within the experimental area of the laying hen experiment (**b**).

**Figure 2 animals-14-00625-f002:**
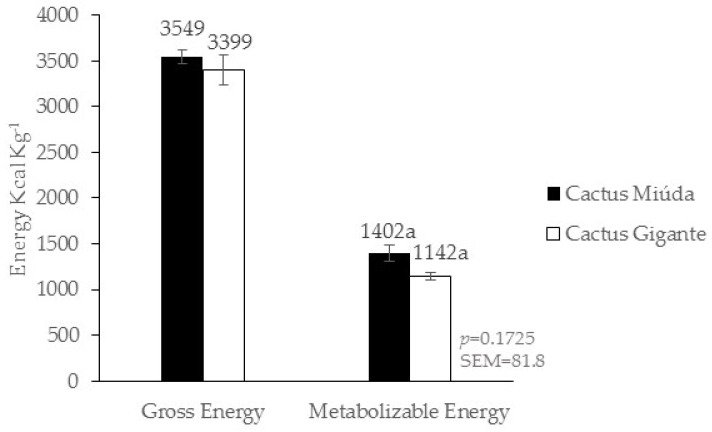
Gross energy and metabolizable energy of the meal of cactus pear varieties Miúda and Gigante for male and female broiler chicks of the Isa Label strain from 24 to 32 days of age (Experiment I). a: black and white colluns (treatments) with same letter indicate statistically no significant differences (*p* ≥ 0.05).

**Figure 3 animals-14-00625-f003:**
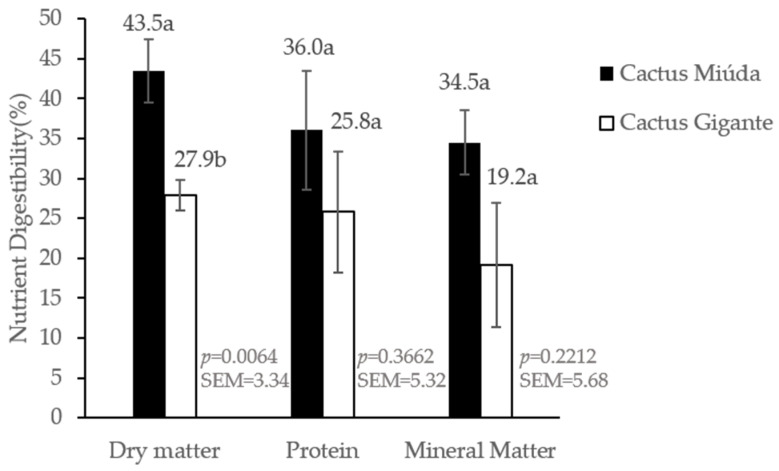
Digestibility coefficient of dry matter, crude protein and mineral matter of cactus pear meal of varieties Miúda and Gigante for male and female broiler chicks of the Isa Label strain from 24 to 32 days of age (Experiment I). a,b: Black and white colluns (treatments) with different letters indicate statistically significant differences (*p* ≤ 0.05).

**Table 1 animals-14-00625-t001:** Composition of the reference diet from the metabolizable energy trial and nutrient digestibility coefficient (Experiment I).

Ingredients	% on as Fed Basis
Corn Grain (8.8% CP)	61.890
Soybean Meal (45%)	32.442
Limestone	0.479
Dicalcium Phosphate	1.046
Common Salt	0.132
Initial Bird premix ^1^	4.000
DL-Methionine	0.009
Total	100.00
**Calculated Composition**	
Calcium (%)	1.160
Clorine (%)	0.150
Bird Metabolizable Energy (kcal/kg)	2.877
Available Phosphorus (%)	0.440
Bird Dig. Lysin (%)	0.954
Bird Dig. Methionine + Cystine (%)	0.700
Crude Protein (%)	20.051
Sodium (%)	0.241

^1^ Guarantee levels per kg of product: calcium (min): 100 g, calcium (max): 200 g/, phosphorus (min): 40 g, methionine (min): 32.276 g, sodium (min): 44 g, iron (min): 600 mg, copper (min): 1600 mg, manganese (min): 1440 mg, zinc (min): 1248 mg, iodine (min): 28.8 mg, selenium (min): 6.6 mg, Vitamin A (min): 140,000 IU, Vitamin D3 (min): 50,000 IU, Vitamin E (min): 260 IU, Vitamin K3 (min): 20 mg, Vitamin B1 (min): 12 mg, Vitamin B2 (min): 110 mg, Vitamin B5 (min): 32mg/kg, Vitamin B12 (min): 240 mgc, Niacin (min): 650 mg, Calcium Pantothenate (min):150 mg, Folic Acid (min): 440 mg, Biotin (min): 0.6 mg, Choline Chloride (min): 4563 mg, Phytase (min): 10,000 FTIU Nicarbazin (min): 2500 mg, Halquinol: 600 mg.

**Table 2 animals-14-00625-t002:** Composition of experimental diets with levels of Miúda CPM in the diet of laying hens (Experiment II).

Ingredients	Miúda CPM Levels (%)			
	0	3	6	9
Corn Grain	65.650	62.650	59.650	56.650
Soybean Meal	16.264	16.914	17.564	18.214
Limestone	8.747	8.498	8.249	7.999
Wheat Bran	5.533	4.372	3.212	2.052
Dicalcium Phosphate	2.141	2.165	2.189	2.213
Soybean Oil	0.546	1.288	2.030	2.772
Common Salt	0.375	0.376	0.377	0.377
Vitini-bird *	0.300	0.300	0.300	0.300
DL-Methionine	0.214	0.218	0.223	0.228
L-Lysin HCl	0.192	0.180	0.168	0.156
L-Threonine	0.025	0.026	0.026	0.027
L-Tryptophan	0.013	0.012	0.011	0.010
Miúda CPM	0.000	3.000	6.000	9.000
Total	100.000	100.000	100.000	100.000
**Energetic and nutritional composition**				
Crude Protein (%)	14.15	14.15	14.14	14.14
Bird Metabolizable Energy (kcal/kg)	2750	2750	2750	2750
Calcium (%)	3.900	3.900	3.900	3.900
Total Phosphorus (%)	0.702	0.692	0.682	0.672
Available Phosphorus (%)	0.500	0.500	0.500	0.500
Sodium (%)	0.160	0.160	0.1600	0.160
Clorine (%)	0.295	0.292	0.289	0.286
Potassium (%)	0.614	0.682	0.750	0.812
Bird Dig. Methionine (%)	0.414	0.416	0.419	0.422
Bird Dig. Methionine + Cystine (%)	0.620	0.620	0.620	0.620
Bird Dig. Lysin (%)	0.750	0.750	0.750	0.750
Bird Dig. Threonine (%)	0.500	0.500	0.5000	0.500
Bird Dig. Tryptophane (%)	0.160	0.160	0.1600	0.160
Linoleic Acid (%)	1.750	2.070	2.391	2.711

* Guarantee levels per kg of product: methionine (min): 160 g, iron (min): 5760 mg, copper (min):1600 mg, manganese (min): 11.52 g, zinc (min): 12 g, iodine (min): 288 mg, selenium (min): 60 mg, Vitamin A (min): 2,000,000 IU, Vitamin D3 (min): 600,000 IU, Vitamin E (min): 5400 IU, Vitamin K3 (min): 300 mg, Vitamin B1 (min): 300 mg, Vitamin B2 (min): 1400 mg, Vitamin B6 (min): 600 mg, Vitamin B12 (min): 4000 mcg, Niacin (min): 6400 mg, Calcium Pantothenate (min): 2600 mg, Folic Acid (min): 400 mg, Biotin (min): 20 mg, Choline Chloride (min): 66 g, Halquinol: 6000 mg.

**Table 3 animals-14-00625-t003:** Chemical composition of Miúda CPM (Experiment II).

Component	Composition *	Component	Composition
Dry Matter (DM)	124.6 g kg^−1^	Dig. Methionine	0.25 g kg^−1^ of DM **
Crude Protein	33.7 g kg^−1^ of DM	Dig. Lysine	1.19 g kg^−1^ of DM **
Ether Extract	14.0 g kg^−1^ of DM	Dig. Threonine	0.52 g kg^−1^ of DM **
Mineral Matter	151.3 g kg^−1^ of DM	Dig. Tryptophane	0.18 g kg^−1^ of DM **
Organic Matter	871.2 g kg^−1^ of DM	Sodium	0.05 g kg^−1^ of DM *
Total Carbohydrates	822.1 g kg^−1^ of DM	Calcium	29.3 g kg^−1^ of DM *
Neutral Detergent Fiber	224.6 g kg^−1^ of DM	Total Phosphorus	0.78 g kg^−1^ of DM *
Acid Detergent Fiber	189.7 g kg^−1^ of DM	Available Phosphorus	0.78 g kg^−1^ of DM

CPM = cactus pear meal, DM = dry matter; * by [6], ** determined by [20] through digestibility coefficient of feeds with fiber content similar to cactus pear of [19].

**Table 4 animals-14-00625-t004:** Climatic Conditions (Experiment II).

Maximum Temperature °C	Minimum Temperature °C	Maximum Humidity %	Minimum Humidity %
39.6	21.9	64.0	13.4

**Table 5 animals-14-00625-t005:** Effect of the use of levels of Miúda CPM on production performance of laying hens (Experiment II).

Variables	Miúda CPM Levels (%)				*p*-Value	*p-*Value Reg	SEM
	0	3	6	9			
EP (%)	89.68	93.95	95.15	93.73	0.3134	ns	1.038
FI (g/bird/day)	114.53	109.82	115.97	109.25	0.6058	ns	2.081
WI (mL/bird/day)	319.41	356.15	348.67	351.03	0.4295	ns	7.948
EW (g)	55.63	53.82	54.19	53.45	0.4142	ns	0.462
EM (g)	49.91	50.55	51.61	50.10	0.8791	ns	0.770
CEM (g/g)	2.31	2.17	2.26	2.18	0.7026	ns	0.043
CED (kg/dozen)	1.54	1.40	1.47	1.40	0.2898	ns	0.028

CPM = cactus pear meal, EP = egg production, FI = feed intake, WI = water intake, EW = egg weight, EM = egg mass, CEM = conversion per egg mass, CED = conversion per egg dozen, reg = regression, SEM = standard error of mean. ns: not significant.

**Table 6 animals-14-00625-t006:** Effect of the use of levels of Miúda CPM on egg quality of laying hens (Experiment II).

Variables	Miúda CPM Levels (%)				*p-*Value	*p-*Value Reg	SEM
	0	3	6	9			
YD (mm)	34.27	33.69	33.50	33.33	0.0887	0.0127 ^L^	0.820
YOLK (%)	24.11	23.63	24.10	23.42	0.1948	ns	0.139
ALB (%)	62.07	62.11	62.61	62.78	0.2431	0.0447 ^L^	0.149
SHELL (%)	10.29	10.41	10.36	10.31	0.8984	ns	0.052
ST (mm)	0.38	0.38	0.38	0.38	0.9599	ns	0.003
SW (g/cm^3^)	1.0922	1.0932	1.0924	1.0927	0.9113	ns	0.000
HU	90.62	93.39	91.54	93.10	0.3660	ns	0.604
SR (g/cm^2^)	4311.81	4254.08	4115.06	3856.69	0.2370	0.0425 ^L^	85.35
YR (g/cm^2^)	17.33	15.65	15.44	14.93	0.7362	ns	1.382
L*	59.49	59.82	60.58	60.37	0.7348	ns	0.358
a*	9.67	8.75	6.99	6.34	0.0004	<0.0001 ^L^	0.378
b*	59.11	52.50	52.53	51.78	0.0266	0.0049 ^L^	1.075

CPM = cactus pear meal, YD = yolk diameter, YOLK % = yolk percentage, ALB % = albumen percentage, SHELL % shell percentage, ST = shell thickness, SW = specific weight, HU = Haugh Unit, SR = shell resistance, YR = yolk resistance, yolk pigmentation parameters L*, a*, and b*. ^L^ = linear, ns = not significant, reg = regression, SEM = standard error of mean.

**Table 7 animals-14-00625-t007:** Effect of the use of levels of Miúda CPM on egg texture profile and yolk pigmentation parameters L*, a*, and b* (Experiment II).

Variables	Miúda CPM Levels (%)				*p*-Value	*p-*Value Reg	SEM
	0	3	6	9			
CI	0.995	0.998	0.996	0.988	0.7857	ns	0.004
HARD	189.5	195.8	186.7	191.0	0.9454	ns	5.114
COHE	0.398	0.380	0.452	0.458	0.0299	0.0122 ^L^	0.012
ELAS	6.0	6.0	6.3	6.5	0.7729	ns	0.076
GUMM	80.8	77.8	86.0	86.1	0.7127	ns	2.992
CHEW	6.1	5.9	6.3	6.2	0.9813	ns	0.409
Col L	81.9	82.1	84.2	84.1	0.4005	ns	0.633
Col a	6.60	5.30	4.26	3.91	0.0054	0.0006 ^L^	0.322
Col b	42.1	34.5	33.1	33.5	0.0001	0.0017 ^Q^	0.847

CPM = cactus pear meal, CI = conversion indicator, HARD = hardness, COHE = cohesiveness, ELAS = elasticity, GUMM = gumminess, CHEW = chewiness, yolk pigmentation parameters L*, a*, and b*, ^L^ = linear, ^Q^ = quadratic, ns = not significant, reg = regression, SEM = standard error of mean.

**Table 8 animals-14-00625-t008:** Effect of the use of levels of Miúda CPM on egg composition, profile of fatty acids, and total cholesterol of yolk of eggs from laying hens (Experiment II).

Miúda CPM Levels (%)	0	3	6	9	*p-*Value	*p-*Value Reg	SEM
**Variables**	**Albumen**						
DM (%)	20.92	17.47	23.16	19.95	0.0061	ns	0.699
MM (%)	2.68	2.41	2.12	2.38	0.0094	0.0202 ^Q^	0.071
CP (%)	9.45	9.22	9.31	9.41	0.8168	ns	0.082
	**Yolk**						
DM (%)	49.9	49.74	49.94	50.51	0.6242	ns	0.282
MM (%)	3.12	3.16	3.16	3.20	0.9976	ns	0.114
CP (%)	3.34	3.75	3.29	3.33	0.3808	ns	0.001
Total FAs (%)	45.94	47.20	45.54	46.22	0.7633	ns	0.516
Myristic (%)	0.486	0.433	0.410	0.392	0.0678	0.0074 ^L^	0.014
Myristoleic (%)	0.079	0.080	0.069	0.070	<0.0001	0.0010 ^L^	0.002
Palmitic (%)	27.51	25.76	25.54	25.18	0.0004	0.0202 ^Q^	0.286
Palmitoleic (%)	4.27	3.21	2.87	2.79	<0.0001	<0.0001 ^Q^	0.178
Stearic (%)	7.58	8.05	8.22	7.99	0.0201	0.0090 ^Q^	0.086
Oleic (%)	39.26	39.46	38.24	37.84	0.0092	0.0028 ^L^	0.237
Vaccenic (%)	2.22	1.97	1.83	1.75	<0.0001	0.0006 ^Q^	0.055
Linoleic (%)	12.38	14.47	16.05	16.79	<0.0001	0.0002 ^Q^	0.512
α-linolenic (%)	0.395	0.517	0.580	0.627	0.0004	<0.0001 ^L^	0.028
Arachidonic (%)	1.701	1.837	1.836	1.806	0.0002	0.0003 ^Q^	0.018
OTHER FAs (%)	4.51	4.86	5.04	5.02	0.0113	0.0036 ^L^	0.075
TC (mg/100 g)	388	469	534	572	<0.0001	<0.0001 ^L^	18.36

CPM = cactus pear meal, DM = dry matter, MM = mineral matter (MM), CP = crude protein, FAs = profile of fatty acids, TC = total cholesterol, ^L^ = linear, ^Q^ = quadratic, ns = not significant, reg = regression, SEM = standard error of mean.

**Table 9 animals-14-00625-t009:** Effect of the use of levels of Miúda CPM on the economic viability parameters of laying hens (Experiment II).

Variables	Miúda CPM Levels %			
	0	3	6	9
Feed intake, kg/bird	7.216	7.150	7.342	7.022
Price/kg of feed, BRL/kg	1.969	1.983	1.997	2.011
Feeding cost, BRL/bird	14.209	14.179	14.664	14.122
Egg dozen, dz/bird	4.708	4.953	5.003	4.943
Feeding cost/Egg dozen, BRL/dz	3.029	2.861	2.932	2.857
Price of egg dozen, BRL	8.000	8.000	8.000	8.000
Gross income, BRL	37.665	39.625	40.021	39.542
Gross margin, BRL	23.458	25.446	25.358	25.420
Relative gross margin, %	100.00	114.97	109.49	109.54

Note: No statistical analysis was conducted for the economic viability parameters presented in this table. CPM = cactus pear meal.

## Data Availability

The data presented in this study are available upon request from the corresponding author. The data are not publicly available due to privacy reasons.

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
