# Peer review of "Use of Cactus Pear Meal in the Feeding of Laying Hens in Semi-Intensive System"

_animals, 2024, doi:10.3390/ani14040625_

Round 1

Reviewer 1 Report

Comments and Suggestions for Authors

1. In abstract, abbreviations must be defined at their first mention, for example, cactus pear meal was first appeared in line 27. Please check the full text.

2. line 77, please define cactus pear meal, then use its abbreviation later in the text.

3. Why did the authors not measure the nutritional value of CPM, but refer to the data published by others? Because the nutritional content of CPM will vary according to the origin and processing technology.

4. The author's experimental design is a little strange. The experimental animals in experiment 1 are broilers, while the experimental animals in experiment 2 are laying hens. Can the results of experiment 1 be applied to experiment 2?

5. why did the authors choose 3%, 6% and 9% CPM? What is the rationale for choosing this dose?

6. Ask the author to check if the units in Figure 2 are mislabeled. “Nutrient digestibility (%)”.

Author Response

Dear Reviewer,

We greatly appreciate your constructive comments and guidance.

We understand the concern regarding the variation in the nutritional content of CPM based on its origin and processing. However, in this study, we chose to reference nutritional data published by researchers from the same institution due to consistency in the origin and cultivation conditions of the cactus. The cactus used in our study was grown at the same experimental station, belonging to the same university and of the same variety as the study referenced. This ensures uniformity in nutritional values. Moreover, the collaboration with researchers involved in the referenced studies, who are part of the same research group, reinforces the reliability of the information used.

We had different purposes in each experiment. As it concerns an innovative food, the first experiment was conducted with the aim of identifying the most suitable forage cactus variety, based on its metabolizable energy value. The second experiment focused on the use of the selected variety, in this case, the best from the first experiment, in the diet of laying hens and its effects on egg production and quality. Regarding the sequential logic of the study, the two-phase approach was a strategic methodological decision to ensure the selection of the best cactus variety before applying it in an egg production context, which is more complex and specific. The choice of the 'Miúda' cactus for the second experiment was based on concrete data obtained in the first experiment, ensuring a scientifically grounded choice. Although both experiments involve birds, they have different objectives and applications, which justifies the difference in the animals used. The first focuses on the energy evaluation of a food, while the second aims to assess the impact of this food on egg production.

To define the 3%, 6% and 9% CPM levels used in our study, we based ourselves on previous research that explored different levels of inclusion of alternative meals in poultry feed. These previous studies, although not specifically focused on cactus meal for layers, provided an overview of different levels of meal inclusion. In one study cited, treatments included up to 40% palm meal for slow-growing chickens, while in the second, palm meal inclusion was tested at up to 20% in diets for Japanese quail. In another study, the inclusion of cactus pear meal varied from 3%, 6%, 9% to 12% in Cobb brand broiler feed. Given these precedents, we decided to explore an intermediate range of CPM inclusion in our study, with the aim of evaluating the response of layers to different levels of inclusion without exceeding the maximum already tested in the literature. We chose 3%, 6%, and 9% to represent low, medium, and high inclusion, respectively, within a range that we believe is safe and potentially effective for layers, based on recommendations and results from previous studies.

Reviewer 2 Report

Comments and Suggestions for Authors

Line 109: it is suggested to write the number seventy-two with digits (72).

Line 112: it is suggested to write the number four with digits (4).

Line 351, 366, 379: as a recommendation, I suggest that you consider revising the tables in your article to include short and precise titles. Additionally, it would be beneficial to provide the meanings of any abbreviations used at the bottom of each table.

Introduction

The introduction highlights the potential of cactus pear meal in the feeding of laying hens. A more explicit statement about the novelty of the study would highlight its contribution to the field.

Materials and methods

This section offers a comprehensive description of the procedures, including the preparation of cactus pear meal and the execution of the experiments. While the general procedures are well-described, adding more specificity in some areas, such as detailed animal care protocols, would enhance the section. Incorporating more visual aids, to summarize experimental setup could enhance reader comprehension. I also suggest redoing the graphs with some more specialized software to make them look better.

Results

The results are detailed, with clear and relevant data on the effects of cactus pear meal in various concentrations.

Discussion

Discussing any potential limitations of the study or alternative interpretations of the results would provide a more balanced view.

Conclusion

Reiterating how the findings address the study´s original objectives could strengthen the conclusion.

Author Response

Dear Reviewer,

We greatly appreciate your constructive comments and guidance.

As suggested, we've changed the spelling of the numbers seventy-two and four to their respective number forms (72 and 4) for clarity and accuracy. We revised all the tables in the article to include shorter, more precise headings. In addition, we have added a caption section at the bottom of each table to clarify all the abbreviations used. Adjustments have been made to the introduction to explicitly highlight the novelty and unique contribution of this study in the field of poultry nutrition, emphasizing the innovative use of cactus meal in laying hens. We've enhanced the section with more details on animal care protocols. We have also incorporated additional visuals to summarize the experimental montage to make it easier for the reader to understand. The graphs have been redone to improve the aesthetics and clarity of the data presentation.

Reviewer 3 Report

Comments and Suggestions for Authors

This study presents a novel investigation into the use of cactus pear meal (CPM) in the diet of laying hens, focusing on production efficiency and cost reduction. This research is significant in the field of poultry nutrition, as it offers a sustainable and economical alternative to traditional feed. The study adhered to ethical protocols and utilized a robust methodology to enhance the reliability of the findings.

Two experiments were conducted: one assessing digestibility and metabolizable energy of two cactus pear varieties and another examining the impact of varying CPM levels on hen performance and egg quality. A comprehensive analysis of feed intake, egg production, egg quality, and composition is commendable. The manuscript is well written, clearly structured, and effectively communicates its findings, appropriately citing relevant literature.

The results indicate that up to 9% CPM in hen diets does not negatively impact performance, but improves egg quality, texture, and nutrition. This suggests that a 3% CPM inclusion offers the most economical benefit without compromising the quality or performance. These findings provide valuable contributions to poultry nutrition and address a critical aspect of sustainable farming.

However, this study has some limitations that should be acknowledged in the Discussion. It focuses only on two cactus pear varieties and does not explore their long-term effects on poultry health and productivity. Further research should include a wider variety of cactus pears and long-term studies to assess their sustained effects on hen health and productivity. Additionally, incorporating parameters for overall hen health and nutritional status would provide a more comprehensive understanding of the effects of CPM.

In conclusion, this study is a significant contribution to the field, highlighting the potential of CPM as a sustainable and cost-effective poultry feed. Future research expanding these findings could further clarify CPM's role of CPM in sustainable poultry farming.

This revision maintains the original intent and content of the review, while streamlining the language and structure for enhanced clarity and precision.

Minor comments:

Dear Authors,

Your manuscript is insightful, and I suggest a minor refinement for improved clarity: consider reviewing the integration of citations to ensure that they smoothly complement the narrative without interrupting the flow. In addition, please ensure consistent adherence to the recommended citation format to enhance readability. These adjustments would further increase the quality of work.

Comments on the Quality of English Language

Examples of minor language and stylistic issues

"Eggs from birds that received CPM levels showed lower yolk diameter values while birds fed corn and soybean meal-based diet had larger yolk diameters but no studies with similar feeds were found for this trait." Issue: This sentence is lengthy and somewhat convoluted, which makes it difficult to follow. For clarity, this can be divided into two sentences.

"The shell strength decreased as the level of CPM increased. Hens fed 9% CPM had lower shell strength when compared to the control feed." Issue: The first sentence is somewhat redundant given the information in the second sentence. This can be condensed into conciseness.

"Myristic and palmitic acids reduced as the level of CPM increased however stearic acid behaved inversely proportional to myristic and palmitic which increase in the yolk as the level of CPM increased in the diet." Issue: Sentence structure is complex and can be simplified. Additionally, there's a lack of punctuation (a comma before "however") that affects readability.

"The cholesterol content in the yolk has become an important issue for consumers as cholesterol is synthesized by the human body and consumers have been advised to avoid dietary cholesterol intake to prevent chronic diseases including coronary heart disease." Issue: This sentence is quite long and can be split into two for better readability. Also, the phrase "including coronary heart disease" might be better placed earlier in the sentence.

These examples illustrate minor stylistic and punctuation issues that, while not drastically affecting overall understanding, could be refined to enhance the clarity and professionalism of the manuscript.

Author Response

Dear Reviewer,

We greatly appreciate your constructive feedback and guidance.

We included the limitations of our study, especially as it relates to the focus on only two varieties of cacti. We also suggest directions for future research, including exploring a wider variety of cacti and long-term studies to evaluate sustained effects, such as exploring the antioxidant potential of CPM. We proofread the entire manuscript to ensure that citations are seamlessly integrated into the narrative without interrupting the flow of the text. As suggested, we've broken up long, complex sentences into smaller parts for clarity. We review punctuation and sentence structure to ensure that the text is concise and readable.

In addition to the specific changes, we conducted a comprehensive review of the manuscript to improve the quality of the language and ensure that the presentation of the results and conclusions is clear and professional. We are confident that these changes have significantly improved the manuscript and welcome the opportunity to further refine our work. We hope that the modifications will meet your expectations and contribute to enriching the value of the study.

Reviewer 4 Report

Comments and Suggestions for Authors

Dear editor,

Thank you for providing me with the opportunity to read this work. Despite being interesting and data-rich, the authors' statistical approach is utterly insufficient for potential publication. Solely relying on analysis of variance, especially in the presence of multiple groups, without being able to identify which one truly differs from the others, lacks scientific validity. The statistical approach is superficial and inadequate for the work to be published

Comments on the Quality of English Language

Minor errors were observed

Author Response

Dear Reviewer,

We appreciate your comments on the statistical approach adopted in our study. We understand your concern regarding the sufficiency of the statistical analysis and would like to clarify our methodology and justify its application.

Our choice to use the Analysis of Variance (ANOVA) followed by the Student-Newman-Keuls (SNK) test for multiple comparisons at a 5% probability level was based on the nature of the data and the objectives of the study. The SNK test is widely accepted and used to identify specific differences between groups when the ANOVA indicates a significant difference. This method allowed us to identify which treatments had significantly different effects from each other. For significant variables, we applied polynomial regression to establish the estimates for the use of cactus pear. This method is suitable for understanding trends and relationships between dependent variables and levels of CPM inclusion, allowing a more in-depth and detailed analysis of the data. We note that our statistical approach is consistent with common practices in the scientific literature, including in papers published in the same journal.

For all analyses, the SAS® University Edition software was used, a robust and recognized tool for data analysis in scientific studies.

We understand the concern, but it was not the interest of the research to initially compare the two cactus varieties. The objective was only to determine the chemical composition and digestibility of the nutrients, which were unknown. However, we understand the observations and agree that this may cause confusion for the reader. Therefore, statistical analysis of the first trial was carried out and the formatting of the graphs was also changed for better understanding.

Round 2

Reviewer 1 Report

Comments and Suggestions for Authors

Can be accepted in present form.

Reviewer 2 Report

Comments and Suggestions for Authors

The authors made the suggested changes and improved the presentation of the manuscript.